# A Lightweight Multi-Level Information Network for Multispectral and Hyperspectral Image Fusion

Mingming Ma [1], Yi Niu [1,2,*], Chang Liu [1], Fu Li [1] and Guangming Shi [1,2]

1   School of Artificial Intelligence, Xidian University, Xi'an 710071, China
2   Peng Cheng Laboratory, Shenzhen 518055, China
*   Correspondence: niuyi@mail.xidian.edu.cn

**Abstract:** The process of fusing the rich spectral information of a low spatial resolution hyperspectral image (LR-HSI) with the spatial information of a high spatial resolution multispectral image (HR-MSI) to obtain an HSI with the spatial resolution of an MSI image is called hyperspectral image fusion (HIF). To reconstruct hyperspectral images at video frame rate, we propose a lightweight multi-level information network (MINet) for multispectral and hyperspectral image fusion. Specifically, we develop a novel lightweight feature fusion model, namely residual constraint block based on global variance fine-tuning (GVF-RCB), to complete the feature extraction and fusion of hyperspectral images. Further, we define a residual activity factor to judge the learning ability of the residual module, thereby verifying the effectiveness of GVF-RCB. In addition, we use cascade cross-level fusion to embed the different spectral bands of the upsampled LR-HSI in a progressive manner to compensate for lost spectral information at different levels and to maintain spatial high frequency information at all times. Experiments on different datasets show that our MINet outperforms the state-of-the-art methods in terms of objective metrics, in particular by requiring only 30% of the running time and 20% of the number of parameters.

**Keywords:** multispectral image; image fusion; multi-level; hyperspectral image

## 1. Introduction

Hyperspectral image (HSI) can be obtained by acquiring the rich spectral information of the scene target as well as the detailed spatial information [1,2], and the spectral and spatial information in its three-dimensional data cube enables the differentiation of different substances in the scene. Accordingly, HSI has outstanding performance for uses in many fields, ranging from remote sensing, change detection [3,4], to land-use classification [5], and other fields such as Digital Soil Mapping (DSM) and Geologic mapping [6,7]. However, it is difficult to obtain 3D hyperspectral data from a single shot using a 2D planar detector [8]. In this condition, a flexible alternative is to acquire both a high spatial resolution low spectral resolution multispectral image (HR-MSI) and a low spatial resolution high spectral resolution hyperspectral image (LR-HSI) of the same static scene, and then combine the advantages of both images to obtain a high resolution (HR) image in both the spatial and spectral domains. The pansharpening technique in remote sensing is closely related to the hyperspectral image fusion problem. This task aims at generating a high resolution multispectral (MS) image from inputs of a high spatial resolution single band panchromatic (PAN) image and a low spatial resolution multi-spectral image. The two different sets of inputs create the two forms of pansharpening concepts—multispectral and panchromatic images, and low-resolution hyperspectral and high-resolution multispectral images different. The PAN image has only one spectral channel, which means it cannot express RGB colors and, on the contrary, the MS image carries a high expression ability of color.

There are many hyperspectral image fusion techniques that have been mentioned in the last two decades. These methods can be roughly divided into two categories depending

on whether deep learning is used or not. The first category is the classical traditional methods, which mainly include the component substitution (CS)-based methods [9–11], multiresolution analysis (MRA)-based methods [12–17] and variational optimization (VO)-based methods [18–20]. The second category is the deep learning methods [21–24] that have become popular over the years.

### 1.1. Traditional Methods

CS-based pansharpening methods have achieved wide application in the field of remote sensing images pansharpening due to their simplicity and efficiency. The general steps of this method are: firstly, the LR-HSI is upsampled and then the result is mapped to other new representation spaces, separating the spectral and structural information of the hyperspectral image in the new transform domain and then replacing the structural information component with a panchromatic image, improving the spatial resolution of the fused image in this way. Representative works of CS-based methods mainly include the principal component analysis (PCA) [25], the Brovey [26], the intensity-hue-saturation (IHS) method [27], and the Gram–Schmidt (GS) method [28]. However, CS-based methods lose part of the spectral information while replacing the spectral image structure information, so the fusion results usually lead to severe spectral distortion and oversharpening.

MRA-based pansharpening methods deal with remote sensing image fusion problems from a spatial perspective. Such methods use pyramid or wavelet transform to decompose LR-HSI and HR-MSI into multiple scales, and then extract the spatial information of HR-MSI of a certain scale and inject it into the corresponding scale of LR-HSI, and finally use the inverse transform to generate the fused image. The decomposition methods used usually include high pass filter (HPF) fusion [29], generalized Laplacian pyramid (GLP) transform [17], wavelet-transform-based [30], and smoothing filter-based intensity modulation (SFIM) [12]. MRA-based methods can solve the spectral distortion problem of the CS-based method to a certain extent. However, in the process of injecting structural information into the panchromatic image at the multi-resolution scale, the fusion result will degrade the spatial information.

CS-based and MRA-based methods both focus on retaining a certain aspect of detailed information, and the fusion results cannot balance structural information and spectral information. VO-based methods regard pansharpening as an ill-posed problem and are build on variational theory. The major process mainly includes two steps: constructing the energy function and designing an optimisation algorithm for the model function. The sparse-based methods [31,32] and the model-based methods [33,34] that evolved from these two steps are two representative VO-related approaches. Since VO-methods obtain optimal solutions with the help of different valid a priori information and effective optimisation algorithms, they can improve the spatial structure information while maintaining spectral fidelity, but it is easy to fall into local optimisation solutions during the optimisation process, resulting in less than optimal final fused image quality results.

### 1.2. DNN-Based Methods

In recent decades, due to the increasing amount of hyperspectral data obtained, the development of theoretical research in deep learning and the increase in computing power have made it possible to process these data [35,36]. In spectral image pansharpening, DL-based methods are gaining widespread attention and recognition for their excellent fusion performance and speed advantages. To be specific, the inputs to the network are specified as LR-HSI and HR-MSI, and the designed network module performs feature extraction and fusion to obtain the desired spectral data cube for comparison with the ground-truth. By making the reconstruction result infinitely close to the real image, a stable network structure can be determined.

Since Huang et al. first applied DL-based methods [37] to remote sensing image fusion, DL-based methods have been promoted and improved in various ways, resulting in many fusion methods based on various types of network architectures. For instance, Masi et al.

applied a super-resolution convolutional neural network (SRCNN) [35] to remote sensing image fusion, and proposed a simple and effective pansharpening neural network (PNN) with a three-layer structure [38]. An improved version of advanced PNN (A-PNN) [39] then optimises network performance and achieves better results. Dong et al. proposed a non-negative structured sparse representation framework based on clustering to mine spatial and spectral correlations [21], and made good progress. To address the problem of inadequate extraction of HR-MSI spatial information with existing deep learning methods, Jiang et al. proposed a convolutional neural network using a differential information mapping strategy to process residual information between HR-MSI and LR-HSI [40]. In addition, Zhao et al. proposed a fusion network based on spatial attention and channel attention mechanisms to improve the network's ability to extract features from input images [41]. On the other hand, Liu et al. used a generative adversarial network (GAN) to synthesize high-quality panchromatic sharpened images for the first time, using a two-stream fusion architecture to generate the desired HR-HSI [42]. To alleviate the dependence on real datasets, unsupervised spectral fusion methods have also received attention [43,44]. Ma et al. proposed a new unsupervised fusion framework [45], where the generator builds an adversarial game with spectral discriminator and spatial discriminator, respectively, to preserve the rich spectrum of hyperspectral images information and spatial information for panchromatic images. A network that learns high-level features by constraining the receptive from increasing in the deep layers was proposed to handle the fusion problem and achieved good results [46].

### 1.3. Motivation and Contribution

The ideal fusion image should have spatial information close to HR-MSI and retain the spectral information of LR-HSI. The DL-based methods improve the traditional fusion methods (CS-based, MRA-based and VO-based) and provide various experiences to deal with the pansharpening problem. Although deep learning has been successful in pansharpening, DL-based methods only focus on fused HR-HSI visual effects and objective evaluation metrics, ignoring the running time issue. To the best of our knowledge, Deep blind HSI fusion network (DBIN+) [47] used a recurrent neural network that cannot be parallelized, with an average forward inference time of 2.17 s. Multispectral and hyperspectral image fusion network (MHF) [48,49] took advantage of the large-scale parallelization of neural networks and relieves the pressure of memory congestion to a certain extent, but it still does not break through the constraints of its network structure, weakening the forward reasoning process, and it still takes 1.67 s to reconstruct a HR-HSI image. A hyperspectral image fusion method based on zero-center residual learning (SpeNet) [50] proposed by Zhu et al. is currently the fastest pansharpening method, requiring only 0.49 s for one forward inference. Although Cheng et al. proposed a multiscale information fusion network for semantic segmentation [51], it uses deeply separable convolution to increase the perceptual field to extract local and global features. This is very different from extracting different levels of spectral image features in a hierarchical manner in the spectral image fusion process. Over the past few decades, many computational spectral imaging systems [52–54] have been proposed that can image in real time, computational spectral imaging does not require scanning operations, it projects the spatially and spectrally encoded 3D data cube on a certain plane, the detector superimposes the aliased images of spatial and spectral information, restores the original data through calculation, and obtains the spatial dimension and spectral dimension at the same time. However, neither traditional fusion algorithms nor DL-based methods can meet the requirements of video frame rate reconstruction, it is still a challenge to build a lightweight feature extraction and fusion network for real-time hyperspectral image synthesis.

This motivates us to design a lightweight multi-level information network (MINet) for multispectral and hyperspectral image fusion to meet the real-time reconstruction requirements of hyperspectral imaging. In particular, our starting point is not to prune the best current pansharpening network to reduce network parameters and running time. We



have analysed the working mechanism of the residual module, which plays a key role in existing multispectral and hyperspectral image fusion networks, and defined a residual activity factor $\eta$ to determine the capability of the residual module used (described in detail in Section 2.1). A lightweight fusion network was then designed based on this principle, in order to increase the correlation between spectral channels, we use multi-level spectral information and feature maps share a hidden layer state in batches, mutually perceive the mean information of each channel map, and complete feature extraction and fusion at the same time. While performing feature extraction, LR-HSI and HR-MSI with different depths are selected for joint-guided detail extraction, which is progressive layer by layer to improve the correlation between spectral images and RGB images and reduce spectral distortion. The main contributions of this paper are summarized as follows.

(1) We developed a lightweight multi-level information network (MINet) for multispectral and hyperspectral image fusion to meet the real-time reconstruction requirements of hyperspectral imaging. The residual information between the different levels of LR-HSI and HR-MSI provides additional spectral detail features that contribute to the fusion results;

(2) We defined the activity factor $\eta$ of residual learning for quantifying the capability of the residual learning module, and compared the proposed network module with the hyperspectral image super-resolution reconstruction method that includes the residual learning module for experiments;

(3) A lightweight residual constraint block based on global variance fine-tuning (GVF-RCB) was designed to extract and fuse the spectral feature information of LR-HSI and the spatial structure information of HR-MSI, which strengthen spatial details and reduce parameters.

The remainder of the paper is organized as follows. Section 2 describes our proposed framework in detail. Section 3 introduces comparative experiments on three datasets and discusses the experimental results. Section 4 draws conclusions.

## 2. Proposed Method

In this section, we will first introduce the proposed residual activity factor, then explain the overall network framework in detail, and finally analyse the working principle of the fully connected beta network.

### 2.1. Residual Activity Factor

Residual learning has proven its effectiveness in image feature extraction and fusion [47,49]. Although fusion networks that include residual blocks have achieved good results, their workings and operational speed still deserve more attention. In fact, using the cross-layer connection strategy used by Dense Block in SpeNet with good performance leads to the increase of network data buffer, which not only increases the computation, but also consumes a large amount of memory in the GPU. Therefore, we first consider the ResNet [55] structure instead of Dense Block to improve the synthesis speed.

Residual learning is one of the most used network structures in deep learning networks, relying on 'shortcut connections' to achieve a very good fit. Formally, denoting the desired underlying mapping as $H(x)$ (an underlying mapping to be fit by a few stacked layers), with $x$ denoting the inputs to the first of these layers. Multiple nonlinear layers can asymptotically approximate a residual function $F(x)$. So the desired underlying mapping $H(x)$ can be expressed as Equation (1).

$$H(x) = x + F(x) \tag{1}$$

We suppose that the residual mapping $F$ to be learned is independent of the number of layers rather considering it as a function $F(x, W_i)$ with $W_i$ is the weight matrix layers for a given block. In general, the residual connection directly outputs the input feature $x$ to $H(x)$, and $F(x)$ learns the tiny residual feature between $x$ and $H(x)$. Compared with the network

structure of the VGG [56] tiled convolutional layer, residual learning can better transfer the features extracted by the previous network layer to the subsequent network layers. Unlike extracting features from a single natural image, the core process of hyperspectral image fusion is the spatial-spectral fusion of a three-dimensional data cube stitched together from upsampled multichannel hyperspectral images and multispectral images (RGB images). Therefore, the problem we are dealing with can be transformed into: extracting the features of the data block formed by the connection of the LR-HSI upsampled with the HR-MSI. The traditional CS-based and MRA-based methods accomplish the task by extracting the spatial information of the multispectral image (HR-MSI) and adding it to the blurred hyperspectral image (LR-HSI). Inspired by this, we analogize the form of residual network, assuming that the upsampled LR-HSI represents $x$ and HR-MSI represents $F(x)$. It is worth noting that HR-MSI is no longer the meaning of residuals. This is a fusion problem of fusing data from two different distributions. Further, we hope that the two kinds of data are in the same order of magnitude as possible as shown Equation (2).

$$\|x\|_1 \approx \|F(x)\|_1, \tag{2}$$

where $\|\cdot\|_1$ is $\ell_1$ norm of a vector, to quantitatively analyze the performance of the residual learning block, we define $\eta$ to denote the activity of residual learning as Equation (3).

$$\eta = \frac{\|x\|_1}{\|F(x)\|_1}. \tag{3}$$

This can be understood as a restriction on LR-HSI and HR-MSI data in the process of feature extraction. Indirectly, it is also possible to determine whether the input and output of the residual block are of one magnitude. As shown in Figure 1, when $\|x\|_1 \approx \|F(x)\|_1$, the structure of the network is similar to the residual module, which extracts hyperspectral information from x and spatial information of the same magnitude from F(x), respectively. When $\|x\|_1 \ll \|F(x)\|_1$, the learning ability of the residual decreases sharply. For the residual module, the network will degenerate into the VGG-Net structure, which is equivalent to considering more spatial information in HR-MSI while ignoring the hyperspectral information in LR-HSI. When $\|x\|_1 \gg \|F(x)\|_1$, the residual learning block degenerates to a constant mapping with infinite learning activity, and the residual fraction does not learn any knowledge and has a learning capacity of 0. At this point, the fusion process did not learn any spatial information from HR-MSI.

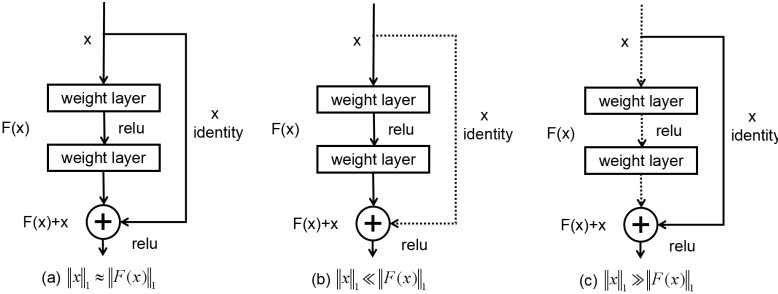

**Figure 1.** Schematic diagram of residual learning degradation. Dashed lines indicate non-existent paths. (**a**) represents $\|x\|_1 \approx \|F(x)\|_1$; (**b**) represents $\|x\|_1 \ll \|F(x)\|_1$; (**c**) represents $\|x\|_1 \gg \|F(x)\|_1$.

According to our definition of the residual activity factor, the residual learning block has an activity value of approximately 1.0 when the magnitudes of the extracted spatial information and inter-spectral information are not particularly significant. In order to verify the effectiveness of the residual activity factor, we use the HARVARD dataset to test three networks with residual modules, DBIN+, MHF-Net and SpeNet, and record the $\eta$ and the corresponding PSNR at different stages in the training process. Figure 2 shows the corresponding PSNR under different $\eta$ by adjusting the distribution of LR-HSI and HR-MSI. At the same time, we also compare the SAM and ERGAS of the three methods as

shown in Table 1. An intuitive conclusion is that the closer the residual activity factor is to 1, the better the performance of the network. Therefore, when evaluating the performance of the residual block in the fusion network, the proposed $\eta$ can be used to evaluate whether the currently used residual block can achieve the desired effect. It is worth noting that $\eta$ is not suitable for evaluating the residual activity in the extraction process of a single natural image, because this is different from the principle that the spectral information and spatial information need to be extracted in the fusion process as close as possible in order of magnitude.

**Table 1.** Comparison of residual activity factors with PSNR, SAM and ERGAS in several super-resolution networks based on residual modules, where $\eta$ is the defined residual activity factor. The best results are shown in bold.

| Method | DBIN+ | MHF-Net | SpeNet |
|---|---|---|---|
| $\eta$ | 0.7407 | 0.8215 | **0.9229** |
| PSNR | 46.20 | 46.59 | **46.69** |
| SAM | 3.8171 | 3.5482 | **3.4380** |
| ERGAS | 0.3398 | 0.3206 | **0.3171** |

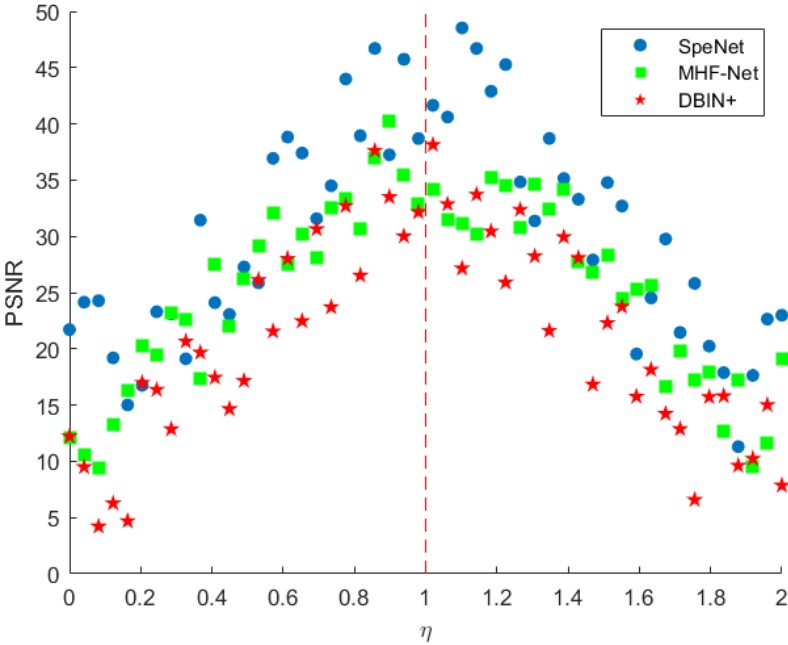

**Figure 2.** The relationship between the normalized PSNR and the active residual activity factor $\eta$ at different stages of the residual learning module.

According to the definition of Equation (3), the value of $\eta$ varies as the input to the neural network changes, but the residual activity factor should be a characterising quantity in a uniform sense. Therefore, for each data $i$ with a sample set of $N$, we count the $\eta_i$ corresponding to each input and calculate the average residual factor $\eta^{'}$ for sample $N$ together reflecting the learning activity of the residuals throughout the network as shown in Equation (4). For the sake of concise representation, $\eta$ appearing in subsequent comparison trials all denote the average result for the overall dataset.

$$\eta^{'} = \frac{1}{N}\sum_{i=1}^{N}\eta_i, \tag{4}$$

where $N$ represents the number of samples. We illustrate the proposed method experimentally in comparison with current hyperspectral super-resolution reconstruction methods in Section 3.4.

### 2.2. Lightweight Multi-Level Information Fusion Network

The proposed lightweight multi-level information network (MINet) for multispectral and hyperspectral image fusion framework is displayed in Figure 3. The network is composed of several residual constraint blocks (orange boxes) based on a global variance fine-tuning (GVF-RCB) module, and a hyperspectral image reconstruction module (green boxes). The input to the network is the LR-HSI $\mathbf{L} \in \mathbb{R}^{h \times w \times B}$ and the HR-MSI $\mathbf{P} \in \mathbb{R}^{H \times W \times b}$, where $(h, w, b)$ are the reduced height, width, and number of spectral bands, respectively, and $(H, W, B)$ are corresponding high-resolution version ($h \ll H, w \ll W, b \ll B$). First, $\mathbf{L}$ is upsampled to $\widetilde{L} \in \mathbb{R}^{H \times W \times B}$ of the same size as $\mathbf{P}$ by bilinear interpolation. In order to reduce the amount of computation and improve the running speed, we choose a bilinear interpolation algorithm that is more suitable for large-scale parallel computation and stable for upsampling operations. Then, the GVF-RCB module estimates a residual image from both the input $\mathbf{P}$ and $\widetilde{L}$ progressively along the spectral dimension. Inspired by the success of the progressive reconstruction strategy in image super-resolution, we progressively embed the spectral bands of upsampled LR-HSI instead of feeding them all into the network at the beginning. The resulting residual image is further superimposed on $\widetilde{L}$ to generated a blurred HR-HSI, which is finally fed into the reconstruction module. After further fine-tuning and correction of the reconstruction module, the network outputs the final reconstructed high-resolution hyperspectral image.

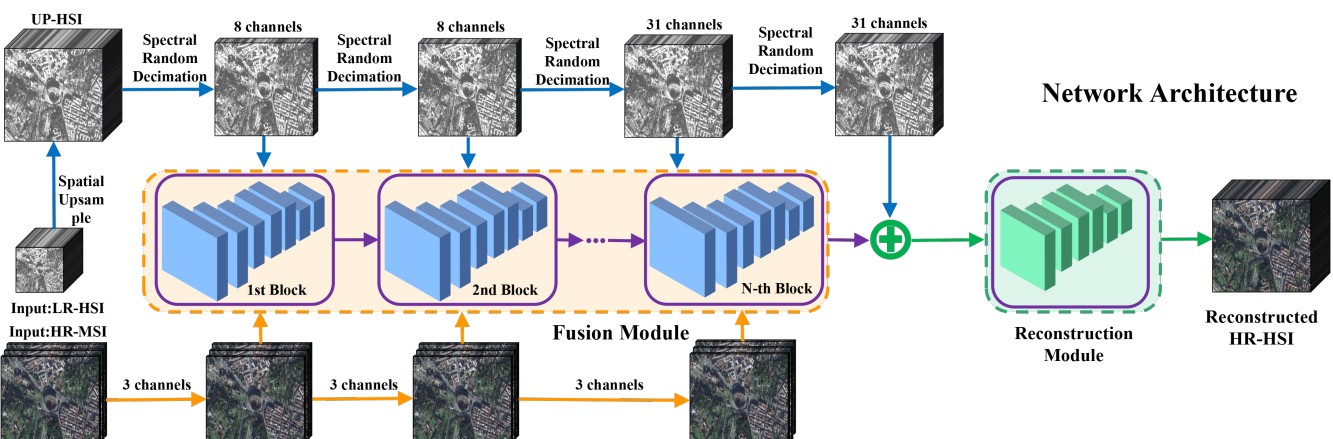

**Figure 3.** The overall flowchart of the proposed MINet for multispectral and hyperspectral image fusion.

The GVF-RCB module is the core module of our proposed multi-level information network, and the flowchart of the network architecture is shown in Figure 4. Its main function is to fuse the spatial information extracted from HR-MSI and the spectral information extracted from LR-HSI to reconstruct the corresponding HR-HSI. The GVF-RCB module is composed of a global residual connection branch and multiple local residual fusion unit cascade branches. The global residual connection branch can extract and transfer low-frequency information of hyperspectral images quickly and efficiently. The goal of residual branch learning is to fuse the information extracted from LR-HSI and MSI into a map of the detail level of the desired HR-HSI to be reconstructed. Define the output of the fusion network as $\mathbf{Z} \in \mathbb{R}^{H \times W \times B}$, which can be expressed as Equation (5).

$$Z = \widetilde{L} + f(\widetilde{L} + P), \tag{5}$$

where $f(\cdot, \cdot)$ represents the mapping of the detailed level of information extracted from LR-HSI and MSI fused into the desired HR-HSI to be reconstructed. The network struc-

ture of the proposed GVF-RCB module mainly uses convolution kernels with receptive field sizes of $3 \times 3$ and $1 \times 1$ as the basic configuration. The $3 \times 3$ convolution kernel is used to extract the spatial feature information and build a fusion model for local spatial information features of hyperspectral images, and the $1 \times 1$ convolution kernel is used to fuse the hyperspectral information for channel compression or dimension enhancement. Inspired by the success of the zero-mean normalization strategy in SpeNet, we also add this constraint to transform the data distribution of feature maps to zero when extracting residual feature information.

The overall network consists of four key steps: (1) concatenate the output of the upper layer with HR-MSI (UP-HSI sampling numbers are 8, 16, 31) and increase the number of channels to be consistent with the up-sampled hyperspectral image (UP-HSI); (2) fuse two kinds of data with different distributions. A fully connected beta network is used to perform mutual induction global spectral information fusion on the output feature map and UP-HSI, adaptively adjust the data distribution of the two inputs, and accelerate the fusion of local spatial information and spectral information in the next stage; (3) use a $3 \times 3$ convolutional layer to fuse the local spatial structure feature information of feature map0 and feature map1, and use a $1 \times 1$ convolutional layer to perform spectral fusion of the channel dimension and reduce the number of channels; (4) use convolution for further fusion, and add it to the feature map and output it.

The GVF-RCB module mainly completes two tasks. First, it extracts the spectral feature information of the hyperspectral image and the spatial structure information of the multispectral image, and fuses the two kinds of information into the detailed information of the hyperspectral image to be output. It is responsible for MINet extraction and fusion of complementary information from each input. In addition, as the basic unit of residual branch information fusion, the spatial structure information and spectral information of different levels are gradually fused through the cascade of multiple GVF-RCB modules. The two fused parts have the same spatial resolution, but the number of channels and spectral images are different for each input to the GVF-RCB module, and the gradually increasing number of channels enhances the feature extraction of spectral information. The two different feature vectors rely on the core unit beta network for fusion.

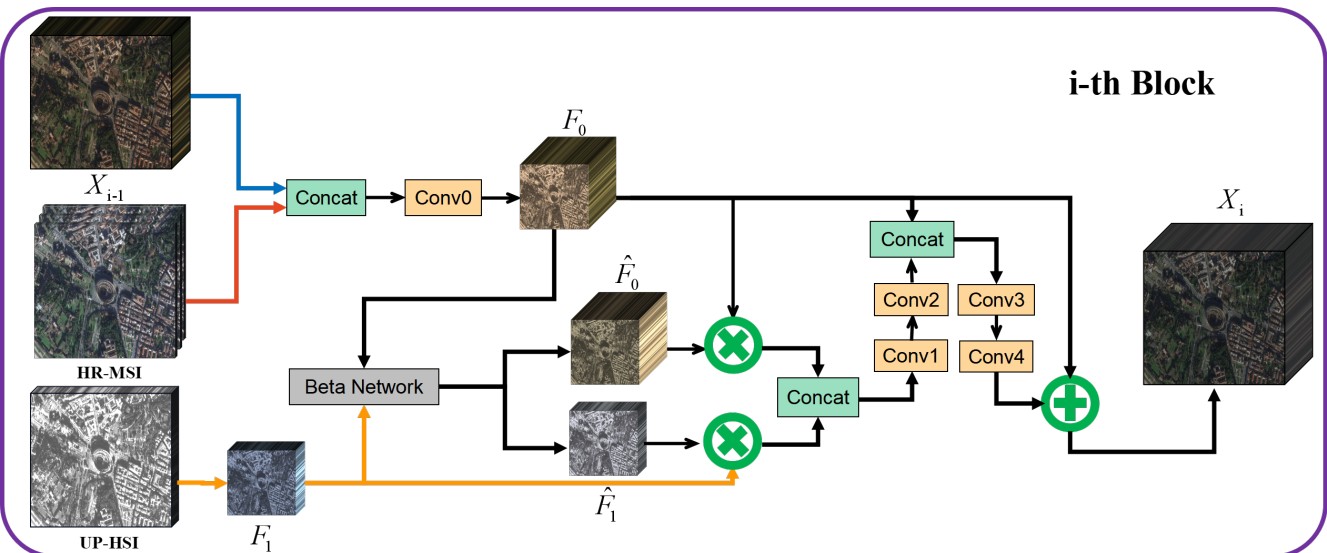

**Figure 4.** The architecture of the lightweight residual constraint block based on global variance fine-tuning (GVF-RCB).

The input of the hyperspectral image reconstruction module is the coarse HR-HSI output by the fusion network, and the reconstructed images in the first stage of the network are fine-tuned and distilled. The network structure of the reconstruction module is relatively

simple, the overall structure is a residual block, and the residual branch is mainly composed of $3 \times 3$ and $1 \times 1$ convolutional layers alternately.

### 2.3. Fully Connected Beta Network

The fully connected beta network is the core unit of the GVF-RCB module, which can be used to adjust the mean and variance of hyperspectral images. Although hyperspectral images and feature maps generated by convolutional neural networks have the same spatial dimension, their data distributions are quite different. We visualized the input and output of the beta network in the fifth GVF-RCB module, and selected three images from the CAVE [57] and HARVARD [58] datasets, respectively, as shown in Figure 5, where blue represents the mean value of each channel of the hyperspectral image and orange represents the mean value of each channel of the feature map. The average value of each channel is calculated as in the average pooling operation. From the mean value of each channel of the hyperspectral image and feature map, and the beta coefficient of each channel (output of the network), it can be seen that the mean value of the hyperspectral image channel is small and densely distributed, while the mean value of each channel of the feature map is large and irregularly distributed. Due to the obvious data gap, directly cascading hyperspectral images and feature maps will result in the ineffective fusion of the spatial feature information of the feature maps and the spectral information of the hyperspectral images. Therefore, it is usually necessary to pass the extracted feature information to convolutional layers using Dense Connection.

Adjusting the internal data distribution of the neural network can often make the network converge quick and stable, and improve the generalization ability of the network. Batch normalization is one of the most commonly used methods to quickly adjust the distribution of feature map data. During training, the local mean and variance of each mini-batch data are used for normalization and to estimate the global value. Finally, in the testing process adjust the data distribution with the global mean and variance instead of the local mean and variance. However, super-resolution reconstruction methods [59] for natural images point out that batch normalization will introduce huge randomness and cause the network to underfit, so it is not suitable for image reconstruction problems.

To reduce the difficulty of training neural networks, we introduce a coefficient $\beta$ ($0 \leq \beta \leq 1$) on the residual learning branch, and the refined residual information is multiplied by $\beta$ to achieve convergence. Assuming $\mathbf{X} \in \mathbb{R}^{H \times W \times B}$ is a feature map, let $\mu$ and $\sigma^2$ be the mean and variance of $\mathbf{X}$, respectively, calculated according to the following equation:

$$\mu = \frac{1}{H \times W \times B} \sum_{i=1}^{H \times W \times B} X_i \tag{6}$$

$$\sigma^2 = \frac{1}{H \times W \times B} \sum_{i=1}^{H \times W \times B} (X_i - \mu)^2, \tag{7}$$

when the output feature map $\mathbf{X}$ of the residual branch is multiplied by a small coefficient $\beta$, the mean $\mu'$ and variance $\sigma'^2$ of the feature map are calculated according to the following equation:

$$\mu' = \frac{1}{H \times W \times B} \sum_{i=1}^{H \times W \times B} (\beta \cdot X_i) = \beta\mu \leq \mu \tag{8}$$

$$\sigma'^2 = \frac{1}{H \times W \times B} \sum_{i=1}^{H \times W \times B} (\beta \cdot X_i - \mu)^2 = \beta^2\sigma^2 \leq \sigma^2. \tag{9}$$

Therefore, multiplying the residual branch by a small coefficient $\beta$ changes the mean and variance of the feature map data, which essentially adjusts the data distribution of the feature map in the direction where the mean tends to zero, reducing the absolute value

of the feature map. Further, the magnitude of the learning parameters in the process of training the network is reduced, so that the variable parameters can better fit the data map. The reduction of variance makes the data distribution more compact, suppresses the fluctuation range of data changes, and at the same time enhances the stability of the data, making the neural network converge towards a more stable direction.

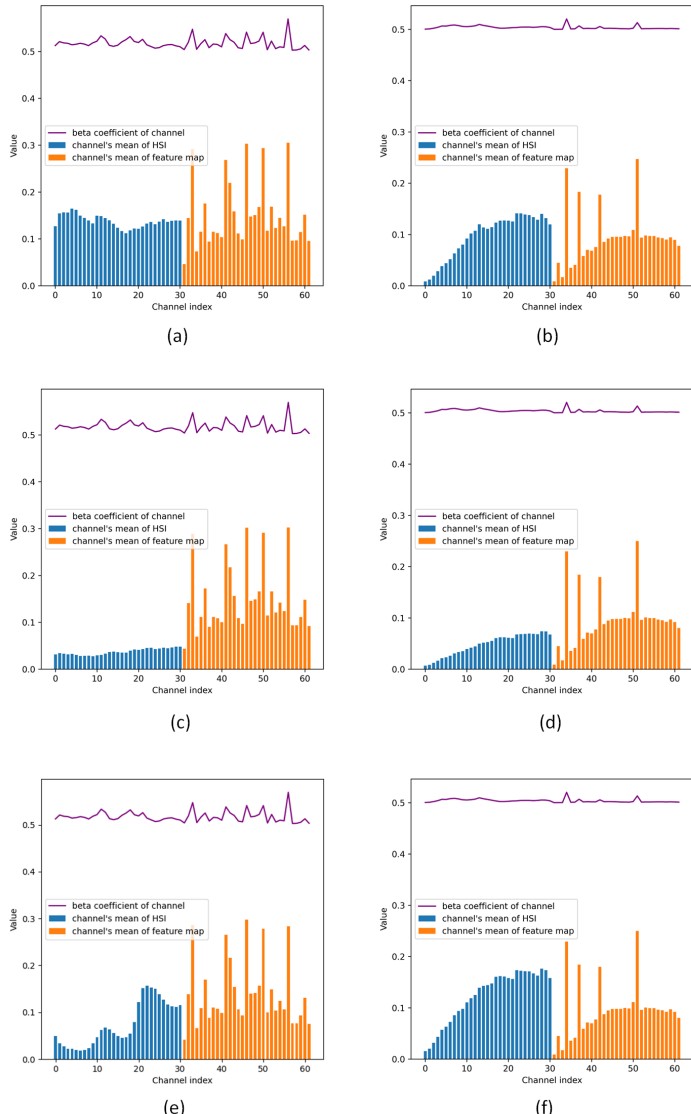

**Figure 5.** Mean and corresponding beta value of hyperspectral image and feature map. (**a,c,e**) represent the data of the three groups of images in the CAVE dataset; (**b,d,f**) represent the data of the three groups of images in the HARVARD dataset.

The proposed fully connected beta network is shown in Figure 6. We used the beta network to make the hyperspectral image and the feature map perceive each other's feature information, and then adaptively changed the data distribution to an appropriate degree. The selection of the beta value was obtained by adaptive learning of the beta network, for each channel will learn a different beta value, and then the small coefficient beta was multiplied by each feature map. The detailed calculation steps of the beta network algorithm are shown in Algorithm 1.

---

**Algorithm 1: Data distribution adjustment in fusion network**

---

**Input:** $X$ and $Y$

**Output:** $X'$ and $Y'$

   *STEP1: compute mean vector*

$$\boldsymbol{\mu}_X \leftarrow \frac{1}{H \times W} \sum_{i=1}^{H \times W} X_{ij}, j = 1, 2, \cdots, C$$

$$\boldsymbol{\mu}_Y \leftarrow \frac{1}{H \times W} \sum_{i=1}^{H \times W} Y_{ij}, j = 1, 2, \cdots, C$$

   *STEP2: concatenate $\boldsymbol{\mu}_X$ and $\boldsymbol{\mu}_Y$ to obtain neural input $\mu$*

   *STEP3: feed $\mu$ into fully connected network and output the coefficient vector $\beta$*

   *STEP4: split $\beta$ to $\boldsymbol{\beta}_X$ and $\boldsymbol{\beta}_Y$*

   *STEP5: compute outputs $X'$ and $Y'$*

$$X' \leftarrow \boldsymbol{\beta}_X^{(i)} * X^{(i)}, i = 1, 2, \ldots C$$

$$Y' \leftarrow \boldsymbol{\beta}_Y^{(i)} * Y^{(i)}, i = 1, 2, \ldots C$$

---

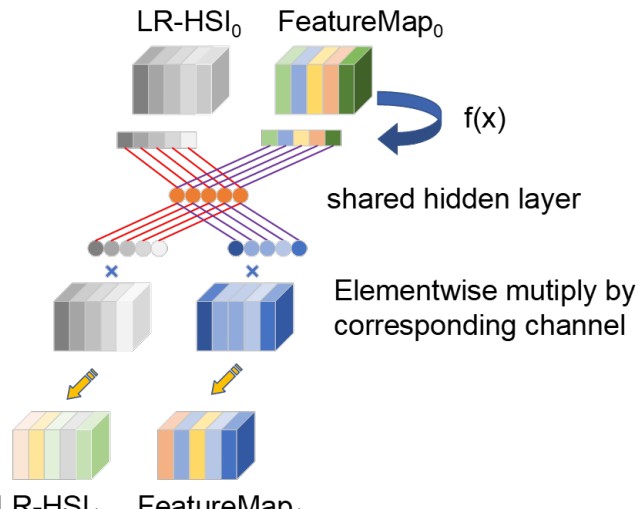

**Figure 6.** The schematic diagram of beta network structure.

## 3. Experiments and Analysis

### 3.1. Experimental Setup

This section is for experimental evaluation. All experiments were run on a server with a Inter(R) Core(TM) i7-6850K CPU 3.60GHZ processor, 32GB of RAM, 2TB disk space, Ubuntu 18.04 and NVIDIA TITAN XP GPUs. We completed the construction of the network model using the PyTorch framework, using a batch size range of $[10, 64]$ during training. We adopted ADAM optimizer [60] to train the network. The initial learning rate of the MINet network was set to $1 \times 10^{-3}$ and the number of iterations was fixed at 3000. For comparison, we selected nine state-of-the-art fusion based HSI SR methods, including coupled sparse tensor factorization (CSTF) [61], non-negative structured sparse representation based method (NSSR) [21], coupled non-negative matrix factorization (CNMF) [62], the low tensor-train rank-based method (LTTR) [63], MHF [48], DBIN+ [47], DMDNet [64], ResTFNet [65], and SpeNet [50].

Seven quantitative metrics that are widely used in multispectral and hyperspectral image fusion tasks were selected to analyse and evaluate the quality of the fused images. First, we used the mean square error (MSE), which is most commonly used in image quality evaluation. It averages the difference between the true and predicted values after a squaring

operation, reflecting the difference between the two sets of images. For each spectral band, the specific formula is shown below.

$$MSE(\widetilde{\mathbf{X}}, \mathbf{X}) = \frac{1}{H \times W \times B} \sum_{H \times W \times B} (\widetilde{\mathbf{X}}_i - \mathbf{X}_i)^2, \tag{10}$$

where $H$, $W$ and $B$ denote the numbers of rows, columns and channels of the image, respectively. $\widetilde{\mathbf{X}}_i$ and $\mathbf{X}_i$ represent the predicted image of the $i$-th spectral band with ground-truth, respectively. The smaller the value of MSE, the closer the predicted image is to the real image.

Peak Signal-to-Noise Ratio (PSNR) is another index for assessing the fusion quality of each band. The definition is based on the MSE and the difference in MSE can be seen by comparing the PSNR. The average PSNR of each band is given by:

$$PSNR(\widetilde{\mathbf{X}}, \mathbf{X}) = -\frac{10}{B} \sum_{k=1}^{B} log(MSE(\widetilde{\mathbf{X}}_k - \mathbf{X}_k)), \tag{11}$$

where $\widetilde{\mathbf{X}}_k$ and $\mathbf{X}_k$ represent the image of the $k$-th channel. Unlike MSE, a higher PSNR value means that the spatial information of the two sets of images is more similar and better between reconstructions.

The relative average spectral error (RASE) estimates the global spectral quality of the pan-sharpened image. It is defined as:

$$RASE(\widetilde{\mathbf{X}}, \mathbf{X}) = \frac{100}{M} \sqrt{\frac{1}{N} \sum_{i=1}^{N} RMSE(B_i)^2}, \tag{12}$$

where $RMSE(B_i)$ is the root mean square error between the $i$-th band of the pan-sharpened image and the $i$-th band of the reference image. $M$ is the mean value of the $N$ spectral bands $(B_1, B_2 \ldots, B_N)$.

Erreur Relative Global Adimensionnelle Synthese (ERGAS) [28] is an error indicator, which represents the effect of pansharpened images from a global perspective. It mainly reflects the spectral distortion of the enhanced image compared to the reference image. The ERGAS index is a normalized dissimilarity index:

$$ERGAS(\widetilde{\mathbf{X}}, \mathbf{X}) = 100\frac{h}{l} \sqrt{\frac{1}{B} \sum_{k=1}^{B} \left( \frac{MSE(k)}{\mu(k)} \right)^2}, \tag{13}$$

where $\frac{h}{l}$ is the ratio of the spatial resolution size of HR-MSI and LR-HSI, and $\mu(k)$ represents the average value in spectral band $B$. The lower ERGAS indicates that the spectral distribution of the enhanced images is similar to that of the ground-truth.

Spectral Angle Mapper (SAM) measures the spectral similarity between the reference image and the fused image by treating the spectrum of each image element as a high-dimensional vector, and by calculating the angle between the two vectors to reflect the likelihood of belonging to the same feature. The SAM index is defined as:

$$SAM(\widetilde{\mathbf{X}}, \mathbf{X}) = \frac{1}{H \times W} \sum_{j=1}^{H \times W} arccos\left( \frac{\widetilde{\mathbf{x}}_j^\mathsf{T} \mathbf{x}_j}{\|\widetilde{\mathbf{x}}_j\|_2 \|\mathbf{x}_j\|_2}, \right) \tag{14}$$

where $\widetilde{\mathbf{x}}_j$ and $\mathbf{x}_j$ are the spectral signatures of the $j$-th ($1 \leq j \leq HW$) pixels of $\widetilde{\mathbf{X}}$ and $\mathbf{X}$, respectively, and $\|\cdot\|_2$ is $\ell_2$ norm of a vector.

Average Structural Similarity Index (ASSIM) represents the proximity of structural information between the ground truth and pansharpened images. The ASSIM index is defined as:

$$ASSIM(\widetilde{\mathbf{X}}, \mathbf{X}) = \frac{1}{B} \sum_{k=1}^{B} SSIM(\mathbf{X}_k, \widetilde{\mathbf{X}}_k), \tag{15}$$

where $SSIM(\cdot, \cdot)$ [66] computes the SSIM value of a typical spectral band. When $SSIM = 1$, the fusion result is the best.

The Correlation Coefficient (CC) is another widely used indicator measuring the spectral quality of the pan-sharpened images. It calculates the correlation coefficient between a pan-sharpened image $\widetilde{\mathbf{X}}$ and the corresponding reference image $\mathbf{X}$ as:

$$CC(\widetilde{\mathbf{X}}, \mathbf{X}) = \frac{\sum_{i=1}^{w} \sum_{j=1}^{h} (\widetilde{\mathbf{X}}_{i,j} - \mu_{\widetilde{X}})(\mathbf{X}_{i,j} - \mu_X)}{\sqrt{\sum_{i=1}^{w} \sum_{j=1}^{h} (\widetilde{\mathbf{X}}_{i,j} - \mu_{\widetilde{X}})^2 \sum_{i=1}^{w} \sum_{j=1}^{h} (\mathbf{X}_{i,j} - \mu_X)^2}}, \tag{16}$$

where $w$ and $h$ are the width and height of the images, $\mu$ indicates mean value of an image. CC ranges from $-1$ to $+1$, and the ideal value is $+1$.

We also use QNR [67], which is composed of spectral distortion index $D_\lambda$ and spatial distortion index $D_s$, as the reference-free measure.

### 3.2. Datasets

To fully evaluate the performance of our proposed MINet network for multispectral and hyperspectral image fusion, we chose two simulated datasets and two real-world hyperspectral datasets for our experiments, including the CAVE dataset, the HARVARD dataset, the National Center for Airborne Laser Mapping (NCALM) and WorldView-2. The following information is specific to the datasets.

(1) The HARVARD [58] dataset consists of 50 indoor and outdoor images captured under daylight illumination. Each image contains $1024 \times 1392$ pixels and 31 spectral bands in a wavelength range of 420 nm to 720 nm. Following [47], in terms of data quantity allocation, we randomly selected 30 images as the training set and the remaining 20 images as the test set.

(2) The CAVE [57] dataset contains 32 images of scenes taken indoors, each hyperspectral image has a spatial resolution of $512 \times 512$ and a spectral range from 400 nm to 700 nm at 10 nm intervals, containing a total of 31 channels. Following [48], 20 images were randomly selected for training and the remaining 12 images were used as the test set in the data allocation process. In order to objectively evaluate the reconstruction performance of each spectral image fusion method, we combine the spectral bands of HSI with the widely used response function of the Nikon D700 camera to generate HR-MSI images of the same scene. Following [47], to obtain LR-HSI, we first filter the image with a $7 \times 7$ Gaussian blur kernel with a mean of 0 and a standard deviation of 2, and then take pixels with step $r$ in both the row and column directions of the spatial dimension of the hyperspectral image.

(3) We selected the remote sensing image dataset released by the Hyperspectral Image Analysis Laboratory of the University of Houston and the National Center for Airborne Laser Mapping (NCALM), which was part of the dataset of the 2018 IEEE GRSS Data Fusion Competition [68] as a pair of real remote sensing image datasets. It contains an RGB image of spatial dimensions 24,040 $\times$ 83,440, which was captured by with the iMAC ULTRALIGHT+ with the focal length of 70 mm, and a 48-band HSI of spatial dimensions 1202 $\times$ 4172, which was captured with the ITRES CASI 1500 with an interval wavelength of 10nm in the range of 380–1050 nm. These sensors are installed on a Piper PA-31-350 Navajo Chieftain aircraft. To obtain ground-truth data, we first intercept HSI images with dimension 1202 $\times$ 4172 and RGB images spatially downsampled to the same dimension as real data. Then we divided the hyperspectral images and RGB images into $2 \times 7$ networks, extracted hyperspectral images and RGB

images with dimensions of $576 \times 576$ from each grid, and used the 7 image patches extracted from the upper half of the hyperspectral images as the training set , and the 7 image patches extracted from the lower half are used as the test set. In order to obtain LR-HSI, we adopted the preprocessing method of the CAVE dataset, and added random additive Gaussian noise with mean 0 and variance 0.0001 to LR-HSI to simulate the low-noise environment of the real environment.

(4) The WV-2 dataset was captured on a commercial satellite and contains an LR-HSI image of size $419 \times 658 \times 8$ and an LR-MSI image (RGB images) of size $1676 \times 2632 \times 3$, while the LR-MSI image is not available. As the ground-truth data are not avaiable, following [48,49], we generated the training data in the following way. To be specific, we select the top half part of LR-HSI ($836 \times 2632 \times 3$) and HR-MSI ($208 \times 656 \times 8$) image to train the MINet and utilize the remaining parts of the dataset as testing data. We first extract $144 \times 144 \times 3$ patches from HR-MSI as network input color images, and $36 \times 36 \times 8$ overlapping patches from LR-HSI as network input low-resolution hyperspectral images, and finally generate training samples.

### 3.3. Experiments and Analysis

We selected two simulated datasets and two real-world hyperspectral datasets for our experiments, using different metrics for comparison. The experimental results of our proposed MINet compared with different methods on the Harvard dataset are shown in Table 2 and bold fonts represent more favorable results. For the fairness principle, the ten methods adopted the same spectral response curve and downsampling process. The significant superiority of our method over state-of-the-art methods is validated. To be specific, the proposed method is only second to DBIN+ on the ASSIM metric, while achieving the best results on PSNR, SAM, ERGAS, Params, FLOPs and GPU memory. It only takes 197.4 ms for MINet to perform a forward inference, which is 12.5% of the time required by MHF-Net, 10% of the time required by DBIN+, and 33% of the time required by SpeNet. In terms of GPU memory consumption, MHF-Net and SpeNet need up to 8 G graphics card memory for inference, DBIN+ needs 4.4 G memory, and MINet only needs 2.5 G to complete the calculation.

The results of the experiments comparing our proposed MINet with different methods on the CAVE dataset are shown in Table 3. All methods were performed using 32× upsampling and in the same test environment. MINet uses 5 GVF-RCB blocks to reduce the number of parameters and run time without reducing the reconstruction effect. The various evaluation metrics demonstrate the superiority of the proposed method, specifically, except that the ASSIM metric is slightly lower than DBIN+, the proposed MINet achieves the best results on PSNR, SAM and ERGAS. Not only have state-of-the-art results been achieved in the various image evaluation indicators, our proposed lightweight network has fewer parameters and faster running time, which can meet the needs of real-time reconstruction.

**Table 2.** Experiment with ten PS methods on the HARVARD dataset. The best performance is shown in bold.

| Method | PSNR (dB) | ASSIM | SAM | ERGAS | Params (M) | FLOPs (T) | Time | GPU Memory (MB) |
|---|---|---|---|---|---|---|---|---|
| NSSR | 34.86 | 0.9312 | 7.1810 | 0.9435 | - | - | 492.5 s | - |
| CNMF | 36.14 | 0.9327 | 5.9471 | 0.9492 | - | - | 200.2 s | - |
| LTTR | 44.32 | 0.9809 | 4.4528 | 0.5223 | - | - | 2316.7 s | - |
| CSTF | 43.64 | 0.9717 | 5.9656 | 0.5943 | - | - | 97.3 s | - |
| MHF-Net | 46.59 | 0.9936 | 3.5482 | 0.3206 | 3.007 | 2.136 | 1647.2 ms | 8087 |
| DBIN+ | 46.20 | **0.9938** | 3.8171 | 0.3398 | 3.083 | 4.125 | 2176.8 ms | 4433 |
| DMDNet | 46.44 | 0.9932 | 3.7562 | 0.3315 | 2.251 | 2.487 | 1456.6 ms | 6847 |
| ResTFNet | 45.86 | 0.9916 | 3.8254 | 0.3396 | 1.525 | 2.365 | 1263.5 ms | 6954 |
| SpeNet | 46.69 | 0.9934 | 3.4380 | 0.3171 | 0.713 | 1.086 | 597.1 ms | 7885 |
| Ours | **46.91** | 0.9936 | **3.3927** | **0.3111** | **0.128** | **0.255** | **197.4 ms** | **2511** |

To more clearly represent the differences in the resulting images, we visualized the residual images on the CAVE data as shown in Figure 7, which takes the ground-truth image as a reference. The images reconstructed by NSSR have local artifacts and noise. The lack of reconstruction of high-frequency details in images by CNMF results in bright foreground contours. Obviously, the spectral fusion method based on deep learning is better than the traditional method based on optimization. The image reconstructed by MINet has less error and the residual map is closer to the ground-truth. In addition, we selected three sets of reconstructed images from the CAVE and HARVARD datasets respectively and compared their PSNR in different spectral bands quantitatively with the comparison method, and the results are shown in Figure 8. It can be seen that the PSNR of MINet is better than that of each comparison method.

**Table 3.** Experiment with ten PS methods on the CAVE dataset. The best performance is shown in bold.

| Method | PSNR (dB) | ASSIM | SAM | ERGAS | Params (M) | FLOPs (G) | Time | GPU Memory (MB) |
|--------|-----------|-------|-----|-------|------------|-----------|------|------------------|
| NSSR | 30.45 | 0.9159 | 10.3595 | 1.5184 | - | - | 117.5 s | - |
| CNMF | 32.44 | 0.8871 | 14.6556 | 1.3654 | - | - | 50.5 s | - |
| LTTR | 41.62 | 0.9706 | 10.8158 | 0.4541 | - | - | 506.5 s | - |
| CSTF | 41.73 | 0.9589 | 12.2554 | 0.4753 | - | - | 19.6 s | - |
| MHF-Net | 45.24 | 0.9932 | 6.9655 | 0.2954 | 3.007 | 534 | 974.6 ms | 5783 |
| DBIN+ | 43.34 | **0.9936** | 7.0630 | 0.3650 | 3.083 | 2063 | 531.6 ms | 1619 |
| DMDNet | 45.75 | 0.9929 | 7.1206 | 0.3245 | 2.251 | 1768 | 654.3 ms | 3517 |
| ResTFNet | 45.12 | 0.9915 | 7.0459 | 0.3369 | 1.525 | 2635 | 362.2 ms | 3268 |
| SpeNet | 46.49 | 0.9931 | 6.7735 | 0.2558 | 0.713 | 271 | 146.0 ms | 6857 |
| Ours | **46.61** | 0.9933 | **6.6853** | **0.2482** | **0.128** | **64 G** | **48.1 ms** | **1169** |

Table 4 shows the results of different methods on the NCALM dataset. The three compared methods decreases significantly and fail to reconstruct high-frequency spatial details, resulting in blurring boundaries. We randomly select a reconstructed image from the dataset, and select the images of the 15th, 33rd, and 44th channels of the reconstructed images of each hyperspectral fusion method for visual comparison. The results are shown in Figure 9. Although DBIN+ achieves high PSNR on the test set, the detail features of the reconstructed image are blurred and have obvious noise. This method cannot effectively suppress the noise contained in the image. The contours of MHF-Net reconstructed images are smoother, but there are still problems of dark spots and blurred details.

Figure 10 shows the comparison results of the proposed method and four methods on the WV-2 dataset. We selected three small areas from the lower right block area of the dataset for detailed magnification and comparison. Our reconstructed high-frequency detail features and smooth parts are better than DBIN+ and MHF, and the Spe method has blurred boundaries. The visualization results show that the proposed method has good visual effect.

**Table 4.** Quantitative performance comparison with the investigated methods on the NCALM dataset. The best results are shown in bold.

| Method | PSNR (dB) | ASSIM | SAM | ERGAS | CC | RASE | QNR |
|--------|-----------|-------|-----|-------|----|------|-----|
| MHF-Net | 39.78 | 0.9789 | 3.8317 | 1.2291 | 0.8561 | 5.4848 | 0.9549 |
| DBIN+ | 39.53 | 0.9756 | 3.8511 | 1.2582 | 0.8462 | 5.9848 | 0.9436 |
| DMDNet | 39.26 | 0.9735 | 3.8621 | 1.2645 | 0.8219 | 5.6847 | 0.9477 |
| ResTFNet | 39.76 | 0.9741 | 3.8456 | 1.2456 | 0.8498 | 5.1678 | 0.9587 |
| SpeNet | 39.96 | 0.9793 | 3.8288 | 1.2222 | 0.8854 | 4.8644 | 0.9650 |
| MINet | **40.22** | **0.9815** | **3.8076** | **1.2187** | **0.8901** | **4.8501** | **0.9654** |

On the whole, the MINet method outperforms existing methods, and the results on the four datasets show that the objective evaluation of the MINet method is well consistent

with the subjective evaluations. In particular, our proposed method operates with the fewest parameters and the fastest speed, highlighting the advantages of light weight.

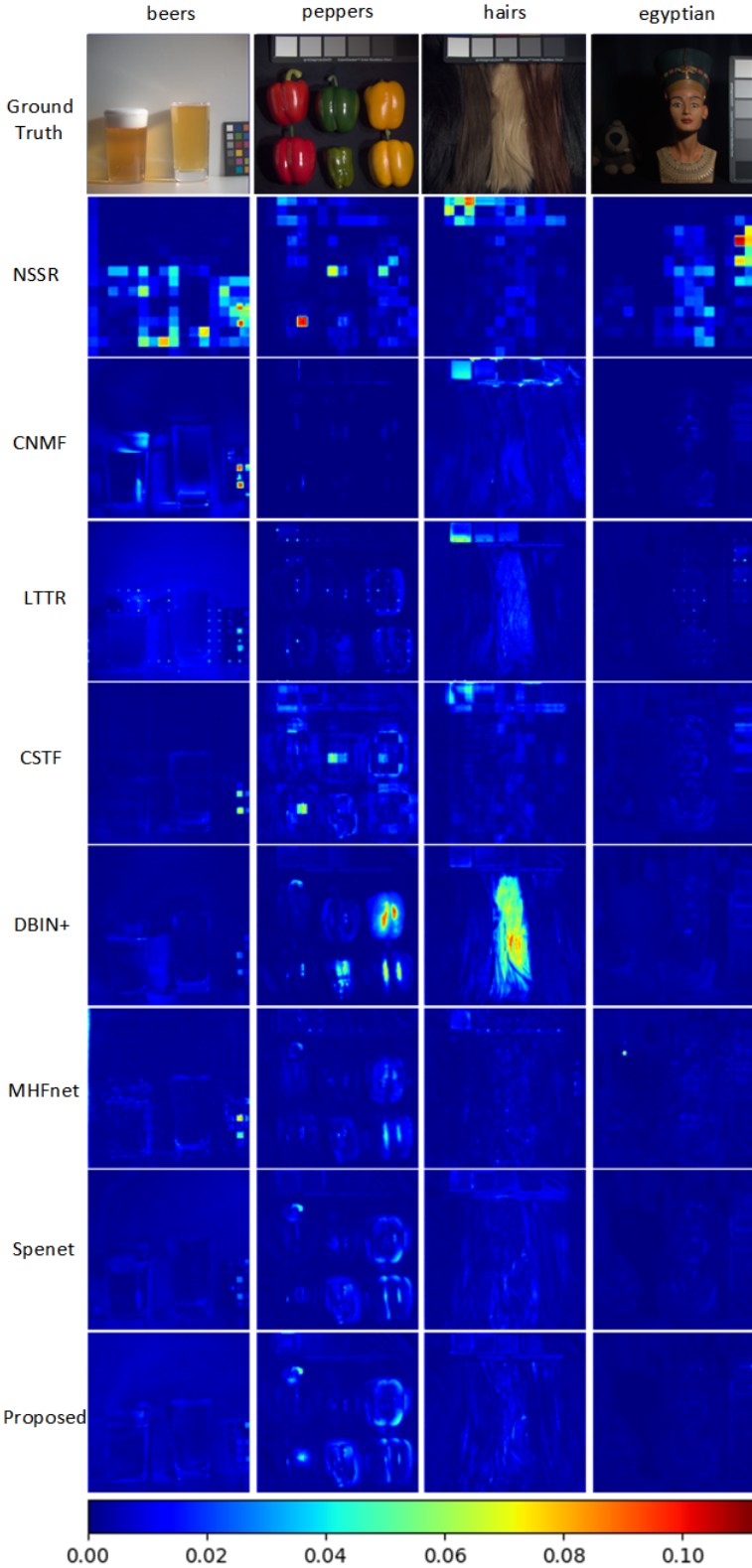

**Figure 7.** Comparisons of the error maps between the spectral bands of reconstructed HR-HSIs.

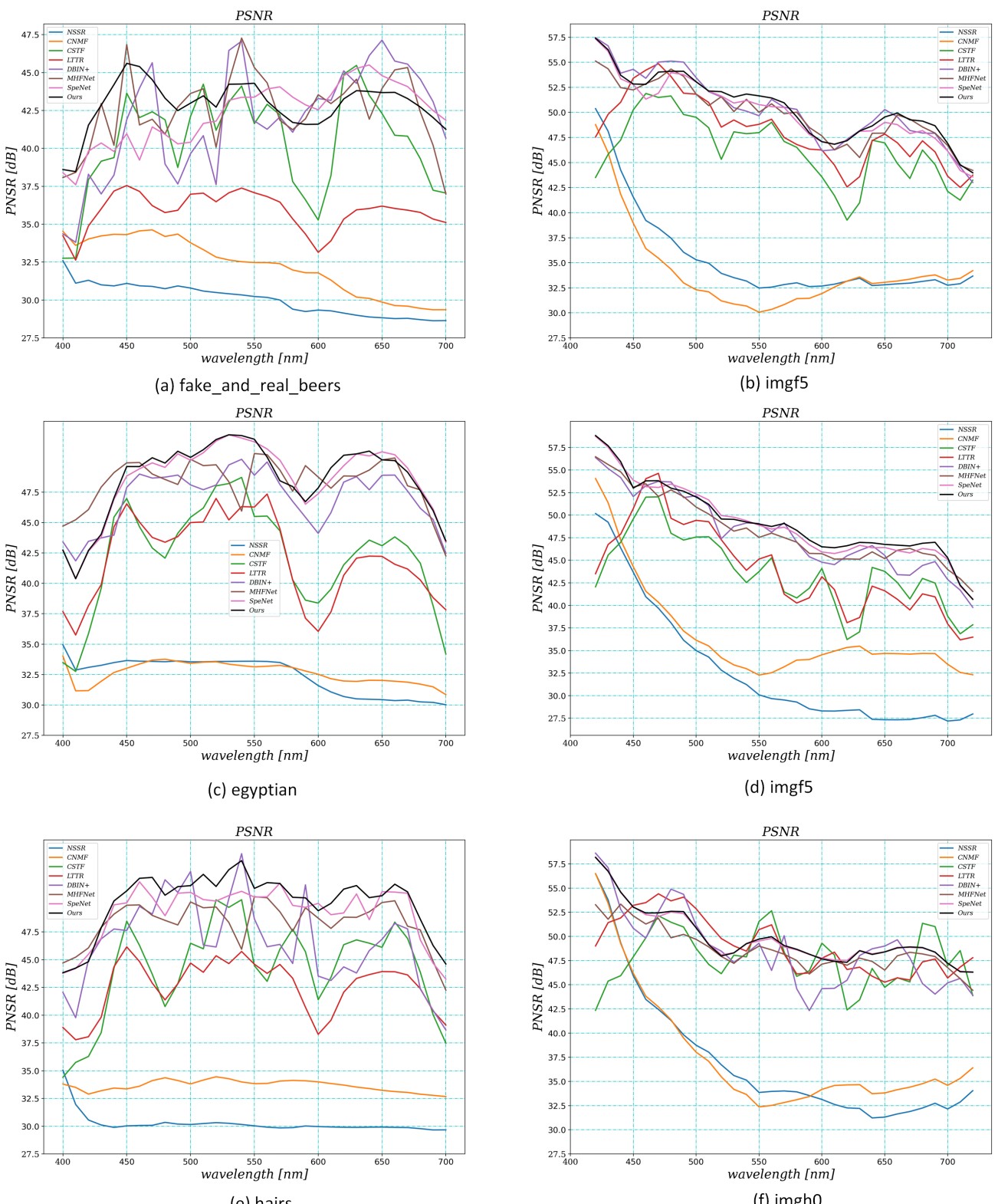

**Figure 8.** Compare PSNR values for each spectral band of the reconstructed HR-HSI. (**a**,**c**,**e**) represent the data of the three groups of images in the CAVE dataset; (**b**,**d**,**f**) represent the data of the three groups of images in the HARVARD dataset.

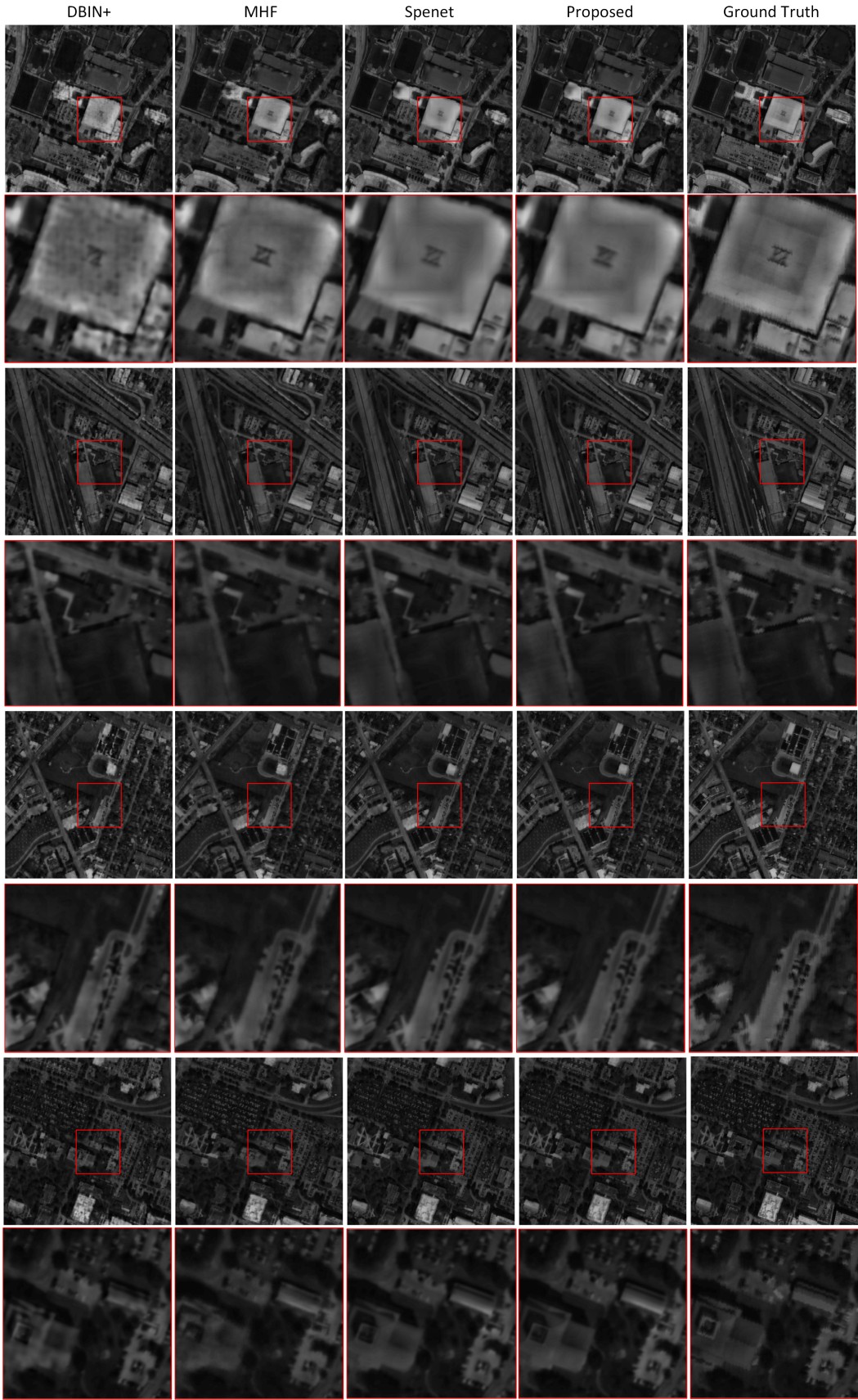

**Figure 9.** The visual comparisons of fusion results obtained by different methods on NCALM dataset.

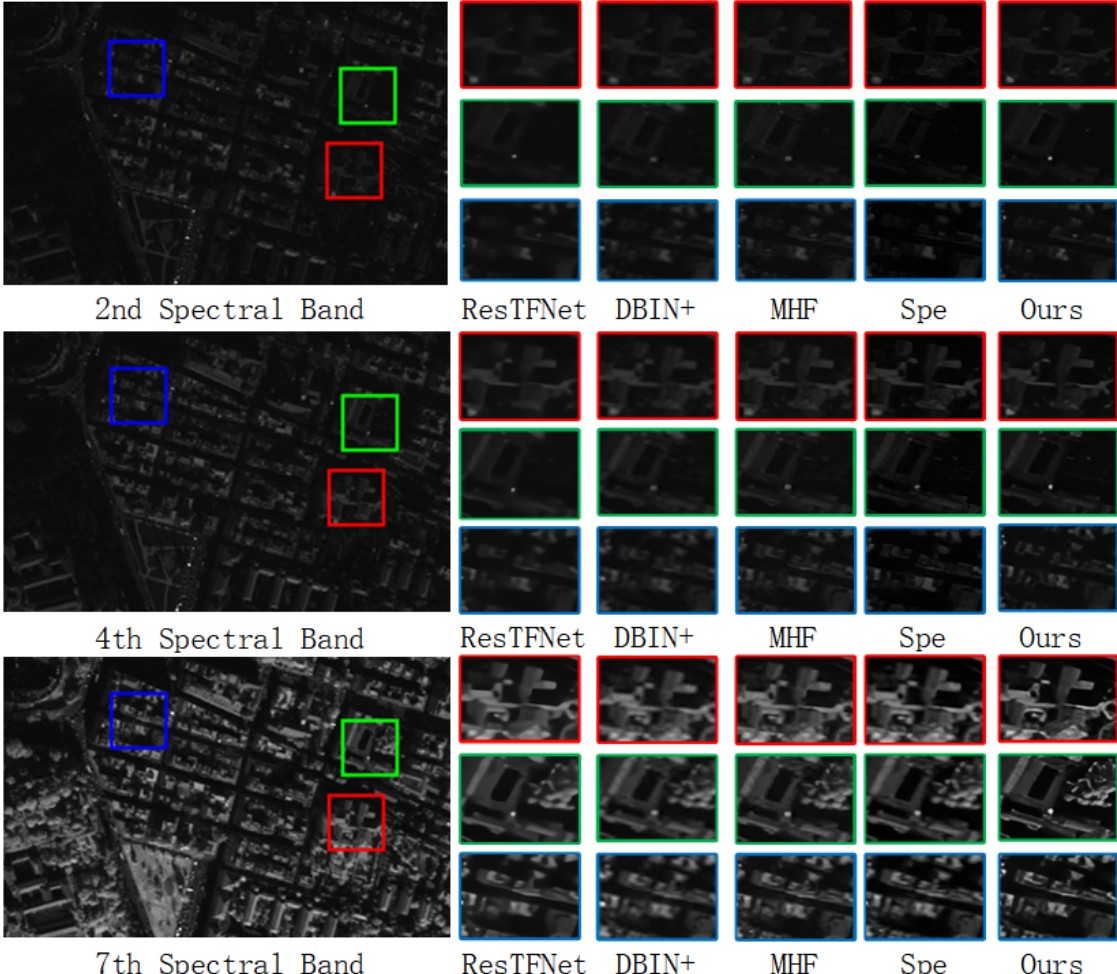

**Figure 10.** The visual comparisons of fusion results obtained by different methods on World View-2(WV-2) dataset.

### 3.4. Ablation Study

In this section, we evaluated the performance of the residual activity factor proposed in Section 2.1 as shown in Table 5. The residual activity value of our proposed MINet on HARVARD data is 0.9873, which is closer to 1 than the state-of-the-art hyperspectral fusion network. Not only is it demonstrated that residual factors can be used to evaluate network activity, but also the superiority of the proposed network is demonstrated. At the same time, we also compared the MINet without the beta network structure to illustrate the performance of the beta network.

**Table 5.** Residual activity comparison with hyperspectral image fusion network on the HARVARD dataset. The best results are shown in bold.

| Method | DBIN+ | MHF-Net | SpeNet | MINet-no-beta | MINet |
|--------|-------|---------|--------|---------------|-------|
| $\eta$ | 0.7404 | 0.8215 | 0.9229 | 0.8957 | **0.9873** |

Then, to verify the performance of the proposed beta network, we compare the performance of MINet and the network after removing the beta network as shown in Table 6. The results of the four evaluation metrics show that the beta network has a positive gain effect on the hyperspectral reconstruction performance. Although the added beta network increases the number of parameters by about 7k, the PSNR is improved by 0.94 dB, and ASSIM, SAM and ERGAS were all significantly improved.

**Table 6.** Contrast effect of beta network in MINet. The best results are shown in bold.

| Network | PSNR (dB) | ASSIM | SAM | ERGAS | Params (K) | FLOPs (G) |
|---|---|---|---|---|---|---|
| MINet-no-beta | 45.67 | 0.9858 | 8.6544 | 0.3278 | **121.704** | **128** |
| MINet | **46.61** | **0.9933** | **6.6853** | **0.2482** | 128.345 | **128** |

## 4. Conclusions

In this paper, we proposed a lightweight multi-level information network (MINet) for multispectral and hyperspectral image fusion to meet the real-time reconstruction requirements of hyperspectral imaging. MINet not only has efficient calculation speed, but also maintains satisfactory accuracy, which is suitable for resource-constrained equipment.

In contrast to the traditional pruning of generalised sharpening networks to obtain lightweight networks, we first analysed the working mechanism of the residual module, which plays a key role in existing hyperspectral fusion networks, and then defined a residual activity factor to quantify the residual learning capability of the module and assess the effectiveness of the proposed module. This inspired us to design a constraint block based on global variance fine-tuning (GVF-RCB) to improve the feature extraction and fusion module. To further improve the efficiency of the fusion network, the proposed network uses multi-level spectral information and feature maps to share a hidden layer state in batches, mutually perceive the mean information of each channel map, and complete feature extraction and fusion simultaneously, simplifying the network complexity while improving the efficiency of feature extraction and information fusion. The ablation experiments also further validate the effectiveness of this module. In addition, we cascade multiple GVF-RCBs in MINet to progressively inherit different levels of feature content into the fused image.

To fully evaluate the performance of our proposed MINet network for pansharpenng, we chose two simulated datasets and two real-world hyperspectral datasets for our experiments. The experimental results demonstrate that the proposed method achieves an average improvement of 0.3 dB in PSNR and achieves state-of-the-art results in most evaluation metrics. More importantly, the network forward inference is three times faster than the best current methods, while reducing the number of parameters by a factor of five. This lightweight design idea and the results validate the effectiveness of the network.

In future work, we will focus on the hardware deployment of the network model, applying it to existing spectral imaging systems and testing it, modifying and debugging the network for specific problems in the engineered system. In addition, we will also test the performance of the residual activity factor on the dataset used in ILSVRC and COCO 2015 competitions.

**Author Contributions:** Conceptualization, M.M., Y.N. and F.L.; methodology, M.M.; software, Y.N.; validation, F.L., C.L. and G.S.; formal analysis, M.M.; investigation, M.M. and Y.N.; resources, G.S.; data curation, C.L. and F.L.; writing—original draft preparation, M.M. and Y.N.; writing—review and editing, C.L. and G.S.; visualization, M.M. and F.L.; supervision, Y.N. and G.S.; project administration, Y.N.; funding acquisition, Y.N. All authors have read and agreed to the published version of the manuscript.

**Funding:** This research was funded by Natural National Science Foundation of China under grants 61875157, 61672404 and 61751310, in part by the National Defense Basic Scientific Research Program of China under grants JCKY2017204B102, in part by Science and Technology Plan of Xi'an under grant 20191122015KYPT011JC013, in part by the Fundamental Research Funds for the Central Universities of China under grants No. RW200141, JC1904 and JX18001, in part by the National Key Research and Development Project of China under grant 2018YFB2202400.

**Institutional Review Board Statement:** Not applicable.

**Informed Consent Statement:** Not applicable.

**Data Availability Statement:** Not applicable.

**Conflicts of Interest:** The authors declare no conflict of interest.

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
