# Peer review of "A Lightweight Multi-Level Information Network for Multispectral and Hyperspectral Image Fusion"

_remotesensing, doi:10.3390/rs14215600_

Round 1
Reviewer 1 Report (New Reviewer)
The manuscript “Lightweight Multi-Level Information Network for Multispectral and Hyperspectral Image Fusion” used residual constraint block based on global variance fine-tuning (GVF-RCB) model to extract and fusion of remote sensing images. In general, it is a well-written manuscript and would be of general interest to the audience of Remote Sensing. However, I think the manuscript could be further improved by adding some in-depth mechanistic discussion. Therefore, my recommendation is moderate revision.
General comments:
- Line 8: « with one-fifth the number of parameters of the state-of-the-art methods », please clarify.
- Lines 28-29 : And other fields such as Digital Soil Mapping (DSM) and Geologic mapping (for example : https://doi.org/10.1016/j.geoderma.2008.06.011).
- Introduction : Pls, discuss the usefulness of remote sensing data fusion in different areas (for example : https://doi.org/10.3390/rs14051103).
- Line 47 : « The second category is the deep learning methods that have become popular over the years. » Pls, add a reference.
- Lines 93 : Please be consistent with the formatting of the reference. The manuscript generally used number-based in-text citations but from this section on, the formatting has changed to include different styles.
- Figure 3. The writing is not clear.
- « 3.3. Experiments and Analysis » Pls compare the obtained results with other studies.
Author Response
Please see the attachment.

Reviewer 2 Report (New Reviewer)
This paper proposed a information fusion framework for multi- and hyper-spectrum images. The topic is very relevant to the journal. The proposed method has novelty in terms of network design to achieve performance comparable to state of the art, with only a fraction of model parameters. The proposed lightweight model is well verified using widely used databases. The paper is in general well written and easy to follow. Also, the concerns raised by reviewers have all been addressed in this revision. So I recommend to accept the paper.Author Response
We would like to thank you very much for your recognition of our work and valuable comments.
Reviewer 3 Report (Previous Reviewer 1)
A brief Summary of the purpose of the paper,
In order to address the needs of real-time reconstruction for hyperspectral imaging, the study suggests designing a lightweight multi-level information fusion network (MIFNet). Based on the residual module activity, a lightweight fusion network was created, and its performance was compared to that of the hyperspectral images super-resolution reconstruction technique. The spectral feature information from LR-HSI and the spatial structure information from HR-MSI is intended to be extracted and fused using the lightweight residual constraint block based on global variance fine-tuning (GVF-RCB). Using data from the National Center for Airborne Laser Mapping, the HARVARD dataset, and the CAVE dataset, the performance of the suggested approach is assessed for fusion. Results obtained from comparing the various evaluations on different datasets show that MINet outperforms better than the state-of-the-art methods.
Comments and Suggestions for Authors
1) These two forms of pansharpening concepts—for multispectral images and panchromatic images, low-resolution hyperspectral images, and high-resolution multispectral images—are distinct, and the Authors should make that clear in the introduction section.
2) The author should discuss the challenges involved in building the suggested lightweight multi-level information network (MINet), for fusing multispectral and hyperspectral images.
3) On Page-11, The author needs to describe how the mean and corresponding beta value of the hyperspectral images and feature map was calculated. Additionally, how is the experimental beta value calculated?
Author Response
Please see the attachment.

This manuscript is a resubmission of an earlier submission. The following is a list of the peer review reports and author responses from that submission.
Round 1
Reviewer 1 Report
The abstract part of the paper should include the author's quantitative findings.
The author ought to identify the tools and software utilized for data analysis.
The conclusion section should be expanded, and the author should make more explanatory connections between the data/method and the conclusions and recommendations.
Reviewer 2 Report
1. Please add DMDNet(Deep Multiscale Detail Networks for Multiband Spectral Image Sharpening) and ResTFNet(Remote Sensing Image Fusion Based on Two-stream Fusion Network) two deep learning based methods in the comparison experiment. Although these are not the latest methods, they are methods that have achieved remarkable results in the field of pansharpening. I hope to see the methods proposed in this paper and their comparison results.
2. The correlation coefficient (CC) is a widely used index for measuring thespectral quality of pansharpened images. And the relative average spectral error (RASE) estimates the overall spectral quality of the pansharpened image. Please compare the experimental results of the different methods on the two indicators.
3. Depending on whether or not reference images are used, they can be divided into reference indicators and non-reference indicators. The results of the full-resolution mode are missed. Please include at least for one sensor.
Reviewer 3 Report
A brief Summary of the aim of the paper,
The paper proposes a to design a lightweight multi-level information fusion network (MIFNet) for remote sensing images pansharpening to meet the real-time reconstruction requirements of hyperspectral imaging. They have analysed the mechanism of the residual module, which plays a key role in existing pansharpening networks, and they defined a residual activity factor to determine its capability. Then a lightweight fusion network was designed based on the residual module activity and is compared with the hyperspectral image super-resolution reconstruction method. The lightweight residual constraint block based on global variance fine-tuning (GVF- RCB) are designed to extract and fuse the spectral feature information of LR-HSI and the spatial structure information of HR-MSI. The performance of the proposed method is fully evaluated for pansharpening, using two simulated datasets and one real-world hyperspectral dataset for experiments, including the CAVE dataset, the HARVARD dataset and the National Center for Airborne Laser Mapping. State- of-the-art results achieved in the comparison of the various evaluations on three datasets validate the superiority of their approach.
I-About the Residual learning method:
- By equation 1 H(x) = x + F(x) where you suppose that the residual mapping F to be learned is independent of the number of layers rather considering it as a function F(x, {Wi}) with Wi is the weigh matrix layers for a given block. This hypothesis must be clarified
- The activity of residual learning as given in equation 2 ∥x∥1 ≈ ∥F(x)∥1 is independent of the Wi and to improve the impact of the performance of the residual learning, the activity η should be tested on the dataset used in ILSVRC & COCO 2015 competitions (see the original paper « Deep Residual Learning for Image Recognition » by Aiming He and al).
- Why only PSNR metric (Peak Signal to Noise Ratio should be added) is compared to η? This comparison should be formally justified, other metrics exist like SAM, RMSE, ERGAS, etc.
- The equation 4 η = 1/N ∑ ηi is fixed and should be discussed.
- MIFNet : is a method published by Jieren Cheng and al . So the reference should be added and your own contribution should to be highlighted and distinguished from the [Origin].
- Why global variance fine-tuning (GVF-RCB) module is chosen? Empirical or formally justification should be added
- P ∈ RH×W×B and not b. The same remark for L ∈ Rh×w×b and not B
- Sentence line 190: « hyperspectral image reconstruction module (orange boxes) »: this should be green boxes.
- as the basic unit of residual branch information fusion, the spatial structure information and spectral information of different levels are gradually fused. So How you can fuse two different features vectors in size and in domain definition?
- Figure 4 illustrated the flowchart of the network architecture and not a simple block (orange box of the figure 3) as written in the figure. Furthermore the section 2.2. line 214 to line 223 describes the hole network. This confusion should be clarified.
- Figure 6 is not clear and not intuitive for interpretation. The beta value seems to vary close to 0.5 for any configuration. How the beta value is chosen experimentaly?
- In line 355, the number of blocks should be added in the table 3.
- Figure 7 should to be changed where ground truth images are not clear and are blue images.
- More real and visual comparisons of fusion results should to be added.
II-About references: -In the related work these references must be added : Hyperspectral Pansharpening Based on Improved Deep Image Prior and Residual Reconstruction, •W. G. C. Bandara, Jeya Maria Jose Valanarasu, Vishal M. Patel Published 6 July 2021, Computer Science, IEEE Transactions on Geoscience and Remote Sensing -[Origin] MIFNet: A lightweight multiscale information fusion network. https://doi.org/10.1002/int.22804 -ResNet [48] and Residual learning [48] have the same reference Et 48 is : He, K.; Zhang, X.; Ren, S.; Sun, J. Deep Residual Learning for Image Recognition. 2016 IEEE Conference on Computer Vision and Pattern Recognition (CVPR), 2016, pp. 770–778. doi:10.1109/CVPR.2016.90.
Automated : Hyperspectral Image Super-Resolution via Non-Negative Structured Sparse Representation
III-About the conclusion: The conclusion should be expanded, and the author should make more explanatory connections between the data/method and the conclusions and recommendations.
Round 2
Reviewer 2 Report
The work in the manuscript is meaningful. It still has some parts of the paper that need to be addressed. Some specifics are given below:
1. For multispectral images and panchromatic images, low resolution hyperspectral images and high-resolution multispectral images, these two types of pansharpening concepts are different and need to be distinguished in your introduction.
2. In line 95, Zhang et al. [37] proposed a fusion network based...., should be "Zhu et al.[37]....". Please check other cites.
3. line 184~line 188, through three experiments, a general curve is given in Figure 2, which is not rigorous enough. One way is to give proof in theory, the other way is to give large-scale experiments, and the experimental data should basically cover different areas on the curve.
4. The resolution of Figure 5 needs to be improved.
5. What is the relationship between residual activity factor and lightweight network? What is its role in designing lightweight networks?
6. The unit of GPU memory consumption is not given in Table 2 and 3.
Reviewer 3 Report
After reading the author's answers, two weak points must be considred :
1) the clarifications added to the new version could be
improved by adding real situations and more experiment
results could be added, In fact authors have used
two simulated datasets and one real-world hyperspectral dataset to evaluate the performance of the proposed MIFNet network for pansharpening. These experiments are not sufficient to generalise the model to any image resolution. How this model could provide relevant features after fusion? How to measure their relevance considering the image Pansharpening problem?
2) NCALM dataset, is the only lidar dataset used in this work to evaluate the method. This dataset is not described (the volume of image used, the complexity of images, ...) in the paper. The title of the paper contains "remote sensing Images Pansharpening ", while the datasets are far from the subject.